# Rehabilitation Interventions Combined with Noninvasive Brain Stimulation on Upper Limb Motor Function in Stroke Patients

**DOI:** 10.3390/brainsci12080994

**Published:** 2022-07-27

**Authors:** Tae-Hyun Cha, Ho-Sung Hwang

**Affiliations:** Department of Occupational Therapy, Konyang University of Occupational Therapy, 158, Gwanjeodong-ro, Seo-gu, Daejeon 35365, Korea; taehyun@konyang.ac.kr

**Keywords:** noninvasive brain stimulation, transcranial direct current stimulation, repetitive transcranial magnetic stimulation, combined rehabilitation treatment, upper extremity rehabilitation, systematic review, medical devices, stroke rehabilitation

## Abstract

(1) Background: This systematic review aimed to focus on the effects of rehabilitation interventions combined with noninvasive brain stimulation on upper limb motor function in stroke patients. (2) Methods: PubMed, MEDLINE, and CINAHL were used for the literature research. Articles were searched using the following terms: “Stroke OR CVA OR cerebrovascular accident” AND “upper limb OR upper extremity” AND “NIBS OR Non-Invasive Brain Stimulation” OR “rTMS” OR “repetitive transcranial magnetic stimulation” OR “tDCS” OR “transcranial direct current stimulation” AND “RCT” OR randomized control trial.” In total, 12 studies were included in the final analysis. (3) Results: Analysis using the Physiotherapy Evidence Database scale for qualitative evaluation of the literature rated eight articles as “excellent” and four as “good.” Combined rehabilitation interventions included robotic therapy, motor imagery using brain–computer interaction, sensory control, occupational therapy, physiotherapy, task-oriented approach, task-oriented mirror therapy, neuromuscular electrical stimulation, and behavior observation therapy. (4) Conclusions: Although it is difficult to estimate the recovery of upper limb motor function in stroke patients treated with noninvasive brain stimulation alone, a combination of a task-oriented approach, occupational therapy, action observation, wrist robot-assisted rehabilitation, and physical therapy can be effective.

## 1. Introduction

Stroke is a temporary or permanent neurological functional disorder resulting from local brain damage caused by a lack of oxygen and glucose supply to the brain for a long period of time because of pathological problems such as bleeding and ischemia in the cerebral vessels [1]. Because of limited upper limb motor function recovery, a total of 25–74% of worldwide stroke survivors need help or are completely dependent on assistance in their daily activities because of functional disorders [2,3]. Neurological damage from stroke occasionally decreases motor cortex excitement, which travels down the spinal cord and reduces motor nerve excitement [4]. In particular, the primary motor area (M1) plays an important role in causing peripheral muscle contractions to make movements, such as reaching [5,6]. Generally, each hemisphere of the brain interacts to balance excitement and inhibition. Reduced M1 excitement on the damaged side stimulates excitement of the corresponding area on the other side, which consequently further reduces the M1 activity on the damaged side [7]. This imbalance between excitement and inhibition has negative effects on upper limb motor function [8].

Intensive rehabilitation is an essential factor in recovery from damage after stroke. However, recently, various studies have attempted to seek ways to make use of direct modulation of brain function as a treatment tool to accelerate the recovery of damaged brain function from neurological diseases rather than using therapies that affect the brain indirectly through physiotherapy [9,10,11]. Among them, noninvasive brain stimulation (NIBS), which is one of the major fields of study in cranial nerve rehabilitation and cognitive science, is utilized in the recovery of motor function in the rehabilitation of stroke patients [12]. Transcranial magnetic stimulation and transcranial electrical stimulation are NIBS, and repetitive transcranial magnetic stimulation (rTMS) and transcranial direct current stimulation (tDCS) are most commonly used in neuroscience and clinical trials [13]. There is a range of NIBS techniques based on their use and their relative advantages and disadvantages [14]. rTMS enables the brain to adapt to environmental and experiential changes through reorganization of the brain based on plasticity [15]. rTMS is a method of generating depolarization of nerve cells in the cerebral cortex by inducing microcurrents in the human brain using magnetic waves, which are generated in a short period of time by placing an electric coil on the outermost skin of the head [16]. rTMS, which is transmitted in a repetitive manner, regulates nerve firing and excites or inhibits brain activity. In healthy volunteers, high-frequency rTMS increased cortical excitability, as measured by a decrease in motor threshold (MT) and an increase in motor evoked potential (MEP) amplitude, whereas low-frequency rTMS inhibited cortical excitability and had the opposite effect on MT and MEP [17,18,19]. Although high-frequency rTMS targeting M1 may improve motor learning of the upper extremities on the opposite side in healthy volunteers, it can reduce motor function of the terminal extremities on the same side [20]. It has been found that low-frequency rTMS improves motor function of the opposite hand through a similar mechanism [21].

In a previous study, task-oriented training after rTMS was effective at relieving upper extremity motor function and stiffness [22]. In addition, combining rTMS and a finger movement program, which sequentially follows instructions, the effectiveness was demonstrated by improvements in hand function [23].

Another NIBS therapy, tDCS, has a positive effect on motor function in the damaged side by inducing neuroplasticity changes in the cerebral cortex caused by changing the excitability of potentials directly in the stimulated part and indirectly in the corresponding part on the other side [7,24]. During tDCS, two electrodes are attached to the scalp, and microcurrents of 1–2 mA are applied, whereby the excitability of the brain nerve is increased under the anodal electrode and decreased under the cathodal electrode [24,25].

tDCS helps restore multiple neurological states by increasing or decreasing cortical excitability in the stimulation region [26]. In stroke patients, many studies have shown that motor function and hand motor tasks can be improved by increasing motor cortical excitability using tDCS [27]. A recent study using functional magnetic resonance imaging (fMRI) reported that motor-related activities increased and motor function improved after using anodal tDCS targeting M1 in a hemisphere with lesions [28]. In addition, a study reported that inhibiting the opposite hemisphere using cathodal tDCS over M1 can improve motor recovery after stroke [29]. A recent study has shown that reducing the excitability of the undamaged hemisphere significantly improves motor learning of paralyzed hands in stroke patients for up to 24 h [30]. In a previous study, the fusion of tDCS and functional electrical stimulation was effective for upper limb motor function [12]. Another study demonstrated the effectiveness of tDCS and virtual reality programs for balance and falls in stroke patients [31].

Previous studies have demonstrated that among the new treatments aimed at improving motor recovery, NIBS techniques such as tDCS and rTMS can induce brain plasticity and are most effective in motor recovery after stroke [32,33]. Recently, studies combining various rehabilitation approaches have been conducted in order to improve functional recovery after stroke [34,35]. Previous studies have shown effective improvement of upper limb motor function with various interventions combined with NIBS [12,22,23,30]. However, the clinical importance of these results are somewhat insignificant, and despite some significant results, two recent systematic reviews have suggested that a lot more information is required to support the use of rTMS and tDCS for stroke recovery [36,37].

Therefore, this study aimed to investigate recent trends and present evidence on the effectiveness of rehabilitation intervention combined with NIBS on upper limb motor function in stroke patients based on academic articles published in the last 10 years. Furthermore, this systematic review of randomized controlled trials (RCTs) investigated the characteristics of the study participants, evaluation tools, application strength and location, application type, and results.

## 2. Methods

### 2.1. Study Design and Literature Research

This study was approved by the Bioethics Committee (KYU-2020-136-01) at Konyang University. This is a systematic review of research methods combining NIBS with various rehabilitation therapies on the basis of RCT in stroke patients among literature published between January 2010 and December 2019. Literature selection was conducted using the Preferred Reporting Items for Systemic Reviews and Meta-Analyses (PRISMA), and the quality of studies and evidence was proven using the Physiotherapy Evidence Database (PEDro) Scale and Patient Intervention Comparison Outcome (PICO).

The literature research was conducted from 23 September to 1 October 2020, using databases including PubMed, MEDLINE, and CINAHL. The titles and abstracts of the studies were reviewed using the following keywords: “Stroke” OR “CVA” OR “cerebrovascular accident”, “upper limb OR upper extremity”, “NIBS” OR “Non-invasive Brain Stimulation” OR “rTMS” OR “repetitive transcranial magnetic stimulation” OR “tDCS” OR “transcranial direct current stimulation”, and “RCT” OR “randomized control trial”. Among the results, 12 studies that met our inclusion and exclusion criteria were selected.

### 2.2. Inclusion and Exclusion Criteria

The inclusion criteria of this study were as follows: studies on stroke patients; studies that reported the effectiveness of rehabilitation therapy combined with NIBS, rTMS, or tDCS; RCT; studies related to upper limb motor function; studies written in English; and studies that allowed full text. The exclusion criteria were as follows: studies on patients with diseases other than stroke; studies only using NIBS, rTMS, or tDCS; studies that excluded upper limb motor function assessment; studies that were not written in English; studies that had limited accessibility; systematic review studies; and meta-analysis studies.

### 2.3. Literature Selection

Literature collection, selection, and quality assessment were independently performed by two researchers. Each of the chosen studies was compared, analyzed, and discussion before a final decision was made. A PRISMA flow diagram was used for study selection (Figure 1). A total of 130 studies were identified, and 31 studies were shortlisted after excluding studies with single-method interventions, reviews, studies contradictory to the subject, and redundant studies. From the 31 studies, 14 were selected, further excluding 17 studies that did not provide body text. Finally, 12 studies were selected after analysis of the body texts, further excluding one protocol research study and one case study.

### 2.4. Quality Evaluation of Literature

In this study, the level of evidence was reviewed using 10 items from the PEDro Scale. The PEDro Scale scores clinical trials based on their reliability and statistical information, and is widely used to evaluate clinical trials [38]. There are a total of 11 items, and “yes” (1 point) or “no” (0 points) is marked, if applicable to each item. The maximum score is 10, which is the sum of the scores of items from 2 to 11, excluding item 1. Scores of 9–10 are considered “excellent”, 6–8 “good”, 4–5 “fair”, and ≤4 “poor”. Thus, the PEDro Scale evaluates the methodological quality of a study.

### 2.5. PICO Evaluation

To determine the effectiveness of rehabilitation therapy combined with NIBS, rTMS, or tDCS, the results of the 12 studies were analyzed using PICO. The types, intensity, location of application, and combined therapy of NIBS therapy were presented as interventional measures.

## 3. Results

### 3.1. Quality Analysis of the Studies

The PEDro analysis revealed that among the 12 selected studies, eight studies were in the “excellent” level consisting of four studies with a score of 10, one study with a score of 9, and three studies with a score of 8. There were four studies in the “good” level, with a score of 6 (Table 1).

### 3.2. Characteristic of Participants

All 12 studies included in this analysis were RCTs. The total number of study participants was 375 in both the experimental and control groups. A total of 241 males and 161 females participated in the study. Five studies included stroke patients who were in their acute phase for <1 year and six studies included those in their acute phase for ≥1 year. The incidence period was unknown in one study. The average age of participants in the 12 studies was 56.57 years. A total of 166 participants had left hemiplegia, and 209 had right hemiplegia (Table 2).

### 3.3. Stimulation Intensity, Frequency, and Duration

tDCS was applied in six studies and rTMS was applied in six studies (Table 3). The stimulation current used in studies with tDCS was 1–2 mA; 1 mA was used in one study, 1.5 mA in one study, and 2 mA in four studies. The application duration was 13–20 min. Among the rTMS studies, five studies used low frequency stimulation at 1 Hz, and one study used high frequency stimulation at 20 Hz. Stimulation intensity was set to 90–120% MT, and 600–1800 pulses were applied for 10–30 min.

### 3.4. Assessment Tools

As an assessment tool to evaluate upper limb motor function, the Fugl−Meyer Assessment (FMA), was most commonly used (11 studies); the Modified Ashworth Scale (MAS) was used in six studies; hand strength was used in five studies; the box and block test (BBT) in three studies; and Wolf Motor Function Test (WMFT) and Brunnstrom stage (BRS) in two studies, respectively. Other assessment tools used were the Barthel Index in three studies and the stroke-specific quality of life in two studies (Table 4).

### 3.5. Combined Physiotherapy

A robotic device was combined with tDCS in two studies, virtual reality in one, motor imagery using a brain–computer interface in one, sensory control in one, and occupational therapy in one. Physiotherapy was combined with rTMS in three studies, and a task-oriented approach, task-oriented mirror therapy, neuromuscular electrical stimulation, and behavior observational therapy were found each in one study, seperately.

## 4. Discussion

This systematic review was performed according to the PRISMA guidelines in order to demonstrate the effectiveness of rehabilitation therapy combined with NIBS therapy on upper limb motor function in stroke patients. NIBS is divided into rTMS and tDCS; it varies depending on the method used and has relative advantages and disadvantages. We conducted a systematic review of 12 RCTs to provide evidence on the impact of rehabilitation therapy on upper limb motor function when combined with NIBS. The quality of the analyzed literature was very high, with eight studies at the “excellent” level. The upper function assessments used to determine the effectiveness of rehabilitation therapy combined with NIBS were FMA, BBT, MAS, Medical Research Council Sum Score, WMFT, range of motion, hand strength, modified Jebsen–Taylor Hand Function Test, Action Research Arm Test, BRS, pinch grip, and manual function test. Among the 12 studies, FMA was the most commonly used (11 studies), followed by MAS and hand strength, which shows that assessment tools for general upper motor function, spasticity, and muscle strength are preferred.

A previous study reported that variables such as electric intensity, current density, and tDCS duration should be mainly taken into consideration for safety when it comes to applying tDCS [50]. In addition, other studies have reported that tDCS has an effect on upper extremity function when combined with other intervention programs [51]. It is important to elucidate which combination of intervention and tDCS would be effective. Two studies used robotic devices for rehabilitation interventions combined with tDCS. These two studies were conducted over 6 weeks. The intensity of tDCS was 2 mA and the action time was 20 min; however, in one study, the effect on upper extremity function could not be confirmed [39]. In contrast, another study found a positive effect on the improvement of upper extremity function and agility [48]. The electrodes in tDCS are typically applied on C3 and C4 of the ipsilesional and contralesional areas, depending on the type of stimulation. The cathodal current decreases the excitability of the brain and the anodal current increases the excitability of the brain. Dual current, in which both currents are stimulated simultaneously, decreases the excitability in the brain of the injured side and increases the excitability in the brain of the normal side. Simultaneous stimulation has been shown to be more effective. In a previous study, the positive effect on upper extremity function was confirmed by combining occupational therapy with tDCS, which applied a bipolar current to the same hemisphere at 2 mA for 20 min [44]. Therefore, combined rehabilitation programs for tDCS and motor skill learning are necessary for the recovery of upper extremity function in stroke patients. Four studies used a tDCS intensity of 2 mA. Among them, an effect was found in three studies, except for a study using the anode and cathode current separately [39,40,41,42]. In a study using an intensity of 1 mA, motor shaping using brain–computer interaction technology was applied to M1 of the injured side, and both the experimental and control groups showed an effect on upper extremity function [45]. In addition, a study using an intensity of 1.5 mA could not confirm a significant difference between the experimental and control groups when applied to bilateral M1 [46]. That is, the upper extremity function is not in the difference in the intensity of tDCS, but in which intervention is applied in combination.

Previous studies have found that the combination of rTMS and upper extremity training leads to positive results [23,52,53,54]. In this study, physical therapies were most frequently combined with rTMS in three cases. In a previous study, the rTMS + PT experimental group and the sham rTMS + PT control group were treated five times a week for 3 weeks for 40 min, but there was no significant difference between the two groups [40]. In another study, three groups of rTMS + PT, rTMS + NMES, and PT were treated for 4 weeks, five times a week, and there was no significant difference between the three groups [47]. However, in a study conducted with two groups, rTMS + PT and sham rTMS + PT, there was a significant difference in wrist stiffness after a total of 10 sessions were performed for 30 min, three times a week [42]. The three studies had the same intensity and frequency at 90% MT and 1 Hz, respectively, and rTMS from 1200 to 1800 pulses was applied. However, the difference in the degree of brain damage in each study participant was found to be a difference in the results. A frequency of ≤1 Hz mostly created an inhibitory effect, and the aftereffect-size decreased as the intensity of rTMS increased, leading to an increased inhibitory effect [55]. In addition, the recovery of upper extremity function was confirmed in two studies that applied task-oriented treatment and action observation [51,53]. Therefore, the combination of an appropriate intensity and an appropriate intervention program can be an effective strategy for relieving spasticity and restoring upper extremity function.

The usefulness of noninvasive brain stimulation for cortical changes in the cerebral hemispheres has been demonstrated in numerous studies [23,50,51,52,53,54]. However, the type of intervention used to improve motor function can be an important factor. In two studies using tDCS that had an effect on upper extremity function, the same stimulation was performed with 2 mA for 20 min [44,48]. However, there was a difference in the stimulation of the ipsilesional M1 and the contralesional M1. The activation of M1 requires a patient-specific identification and approach, rather than a general protocol. For this, fMRI should be used to confirm the damaged state. In addition, three studies using rTMS that were effective in upper extremity function rehabilitation demonstrated that stimulation of the contralesional primary motor cortex using a low frequency of 1 Hz was able to restore upper extremity function and relieve stiffness [39,51,53]. However, MAS, which evaluates stiffness, has the subjective meaning of the evaluator; therefore, it seems difficult to make an accurate evaluation. Future studies should use tools that can more scientifically prove the evaluation of spasticity. Using rTMS, it was confirmed that excitability was altered and regulated in the cerebral cortex, but the optimal intensity could not be confirmed through the analyzed studies [39,51,53]. Further studies are required to elucidate this aspect. This study confirmed that noninvasive M1 stimulation should be used according to the patient’s individual characteristics in order to activate upper limb motor function. This can have different effects, even if the same NIBS protocol is used [44]. In addition, appropriate interventions should be used in combination to maximize the effects on upper limb motor function. Effective interventions induced voluntary movements in stroke patients.

In this study, rehabilitation interventions combined with rTMS and tDCS were compared to determine the effectiveness of rehabilitation interventions combined with NIBS. Recently, NIBS has been actively researched in the rehabilitation intervention process for stroke patients [12]. Among the 12 studies we reviewed, the interventions that were effective for upper limb motor function recovery were the task-oriented approach, occupational therapy, action observation, and wrist robot-assisted rehabilitation, and the intervention that was effective at relieving spasticity was physical therapy. However, in the process of analysis, it was difficult to determine the most appropriate brain stimulation method, because each study had different methods. Additionally, it was difficult to prove the effectiveness of a single treatment method because this study compared brain stimulation and fused rehabilitation treatments.

Study designs will be more accurate if they are conducted with a consistent setting for the control group in the future. Further studies investigating various rehabilitation therapies combined with NIBS for functional recovery after stroke should be conducted.

## 5. Conclusions

Herein, we systematically reviewed rehabilitation therapies combined with NIBS published during the last 10 years in an attempt to determine the effectiveness of rehabilitation therapies combined with NIBS on upper limb motor function in stroke patients. FMA was the most commonly used assessment tool for evaluating upper extremity function in stroke patients. The task-oriented approach and occupational therapy were effective rehabilitation therapies combined with tDCS and rTMS. This result provides useful evidence for rehabilitation treatment interventions combined with NIBS in stroke patients.

## Figures and Tables

**Figure 1 brainsci-12-00994-f001:**
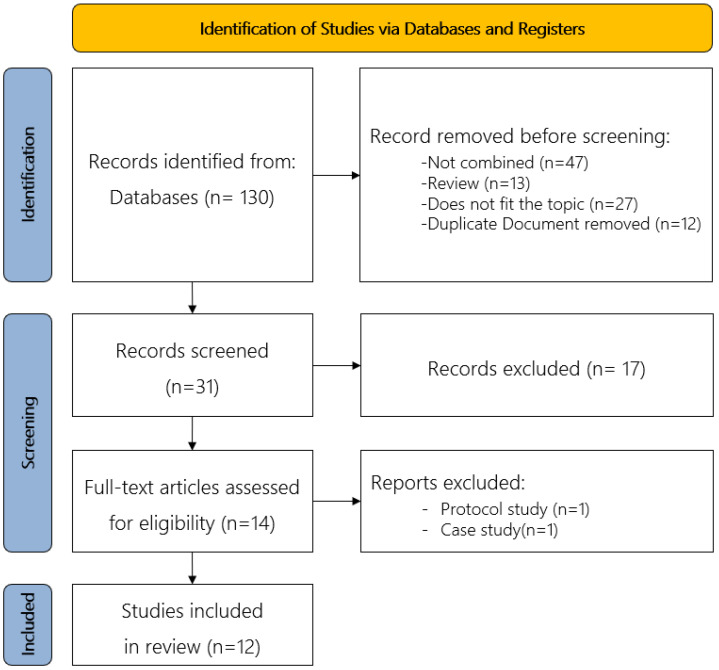
PRISMA flow diagram.

**Table 1 brainsci-12-00994-t001:** PEDro scale for research.

No.	Author (Year)	1	2	3	4	5	6	7	8	9	10	11	Total	Quality
1	Hesse (2011) [39]	Y	Y	Y	Y	Y	Y	Y	N	N	N	N	6	Good
2	Seniów (2012) [40]	Y	Y	Y	Y	Y	Y	Y	N	N	N	N	6	Good
3	Wang (2012) [41]	Y	Y	Y	Y	Y	Y	Y	Y	Y	Y	Y	10	Excellent
4	Barros Galvão (2014) [42]	Y	Y	Y	Y	Y	Y	Y	Y	Y	Y	Y	10	Excellent
5	Viana (2014) [43]	Y	Y	Y	Y	Y	Y	Y	Y	Y	Y	Y	10	Excellent
6	Ilić (2016) [44]	Y	Y	Y	Y	Y	Y	Y	Y	Y	Y	Y	10	Excellent
7	Hong (2017) [45]	Y	Y	Y	Y	Y	Y	Y	N	N	N	N	6	Good
8	Koh (2017) [46]	Y	Y	Y	Y	Y	Y	Y	N	N	N	N	6	Good
9	Tosun (2017) [47]	Y	Y	Y	Y	Y	Y	Y	Y	Y	N	Y	9	Excellent
10	Kim (2018) [22]	Y	Y	Y	Y	N	N	Y	Y	Y	Y	Y	8	Excellent
11	Mazzoleni (2019) [48]	Y	Y	Y	Y	Y	N	N	Y	Y	Y	Y	8	Excellent
12	Noh (2019) [49]	Y	Y	Y	Y	Y	N	N	Y	Y	Y	Y	8	Excellent

**Table 2 brainsci-12-00994-t002:** General characteristic of the reviewed studies.

No.	Author (Year)	Participant (M/F)		Age (Mean)		Hemiparesis (Lt./Rt.)		Duration (Months)	
		EG	CG	EG	CG	EG	CG	EG	CG
1	Hesse (2011) [39]	20/12 ^a^ 18/14 ^b^	21/11	63.9 ^a^ 65.4 ^b^	65.6	18/14 ^a^ 17/15 ^b^	16/16	0.79 ^a^ 0.88 ^b^	0.88
2	Seniów (2012) [40]	12/8	14/6	63.5	63.4	10/10	7/13	1.39	1.26
3	Wang (2012) [41]	8/4	7/5	62.9	64.9	8/4	6/6	24.3	22.3
4	Barros Galvão (2014) [42]	6/4	7/3	57.4	64.6	3/7	7/3	47.8	58.9
5	Viana (2014) [43]	9/1	7/3	56.0	55.0	1/9	0/10	31.9	35.0
6	Ilić (2016) [44]	10/4	7/5	58.3	62.0	1/13	1/11	41.0	37.3
7	Hong (2017) [45]	5/4	8/1	52.7	56.4	1/8	1/8	33.8	33.3
8	Koh (2017) [46]	8/6	7/4	55.3	56.9	8/6	6/5	15.8	13.4
9	Tosun (2017) [47]	6/3 ^a^ 4/3 ^b^	5/4	57.6 ^a^ 56.0 ^b^	61.3	3/6 ^a^ 4/3 ^b^	5/4	1.64 ^a^ 1.98 ^b^	1.57
10	Kim (2018) [22]	4/4	4/8	51.0	74.1	2/6	5/7	1.63	1.75
11	Mazzoleni (2019) [48]	8/12	7/12	67.5	68.7	9/11	8/11	NI	NI
12	Noh (2019) [49]	4/7	6/5	66.4	57.4	6/5	6/5	1.17	0.75

^a^: Experimental Group 1, ^b^: Experimental Group 2, CG: Control Group, EG: Experimental Group, F; Female, M: Male, NI: No information available.

**Table 3 brainsci-12-00994-t003:** Intervention overview.

No.	References	Intervention			
		Type	Intensity/Frequency/ Pulse/Duration	Positions of the Electrodes	Combined Therapy
1	Hesse et al. (2011) [36]	tDCS	2 mA/# #/20 min	Hand area of ipsilesional M1 ^a^ Hand area of contralesional M1 ^b^	a-tDCS + AT ^a^ c-tDCS + AT ^b^
2	Seniów et al. (2012) [50]	rTMS	90% MT/1 Hz/ 1800 pulse/30 min	Contralesional M1	rTMS + PT
3	Wang et al. (2012) [51]	rTMS	90% MT/1 Hz/ 600 pulse/10 min	Contralesional M1	rTMS + TOT
4	Barros Galvão et al. (2014) [39]	rTMS	90% MT/1 Hz/ 1500 pulse/#	Contralesional M1	rTMS + PT
5	Viana et al. (2014) [52]	tDCS	2 mA/# #/13 min	M1	tDCS + VR
6	Ilić et al. (2016) [38]	tDCS	2 mA/# #/20 min	Ipsilesional M1	a-tDCS + OT
7	Hong et al. (2017) [48]	tDCS	1 mA/# #/20 min	Ipsilesional M1	tDCS + MI-BCI
8	Koh et al. (2017) [44]	tDCS	1.5 mA/# #/#	Bilateral primary motor cortex (M1)	tDCS + SM
9	Tosun et al. (2017) [45]	rTMS	90% MT/1 Hz/ 1200 pulse/20 min	M1	rTMS + PT ^a^ rTMS + NMES ^b^
10	Kim et al. (2017) [19]	rTMS	90% MT/20 Hz/ 1500 pulse/15 min	M1	HFrTMS + TOMT
11	Mazzoleni et al. (2019) [37]	tDCS	2 mA/# #/20 min	M1	tDCS + Wrist RAR
12	Noh et al. (2019) [53]	rTMS	120% MT/1 Hz/ #/20 min	Contralesional M1	rTMS + AO

^a^: Experimental Group 1, ^b^: Experimental Group 2, tDCS: transcranial direct current stimulation, rTMS: repetitive transcranial magnetic stimulation, M1: primary motor cortex, a-tDCS: anodal transcranial direct current stimulation, c-tDCS: cathodal transcranial direct current stimulation, AT: arm robot, PT: physical therapy, TOT: task-oriented treatment, VR: virtual reality, OT: occupational therapy, MI-BCI: Brain-computer interface-assisted motor imagery, SM: Sensory Modulation, NMES: neuromuscular electrical stimulation, TOMT: task-oriented mirror therapy, Wrist RAR: Wrist Robot-Assisted Rehabilitation, AO: action observation.

**Table 4 brainsci-12-00994-t004:** Study specific results.

No.	Outcome	
	Time of Intervention	Assessment and Result (EG1/(EG2)/CG)
1	40 min/d, 6 weeks	FMA (−/−/−)/BBT (−/−/−)/MAS (−/−/−)/MRC (−/−/−)/BI (−/−/−)
2	40 min/d, 5 d/w, 3 weeks	WMFT-FAS (+/+)/WMFT-TIME (−/+)/FMA (+/+)/NIHSS (+/+)
3	30 min/d, 10 times	FMA^+^/MEP^+^
4	30 min/d, 3 d/w, 10 times	Wrist MAS+ (+/−)/UL-FMA (+/+)/FIM (+/−)/Wrist ROM (+/−)/SSQOL (−/+)
5	3 d/w, 5 weeks, 15 times	UL-FMA (+/+)/WMFT-TIME (+/+)/WMFT-FAS (+/+)/MAS (−/−)/ Hand strength (+/+)/ SSQOL (+/+)/SSQOL-UL+ (−/−)
6	45 min/d, 2 weeks, 10 times	mJTHFT (+/−)/UL-FMA (−/−)/Hand strength (−/−)
7	40 min/d, 2 weeks	CBF (+/+)/FMA (+/+)
8	30 min/d, 3 d/w, 8 weeks	UL-FMA^−^/MAS^−^/ARAT^−^/BI^−^
9	5 d/w, 4 weeks, 20 times	BRS (+/+/+)/FMA (+/+/+)/BI (+/+/+)/MAS (−/−/−)
10	5 d/w, 2 weeks	Hand strength (+/+)/Pinch grip (+/+)/BBT (+/+)
11	5 d/w, 6 weeks	FMA (+/+)/MAS (−/−)/MI (+/+)/BBT (+/−)
12	1 h BID, 5 d/w	BRS (−/+)/FMA (+/+)/MFT distal (+/−)/Hand strength (+/−)

^+^: Significant differences between groups, ^−^: No significant differences between groups +: Significant changes before and after the experiment, −: No significant changes before and after the experiment, FMA: Fugl-Meyer Assessment, BBT: Box and block test, MAS: Modified Ashworth Scale, MRC: Medical Research Council Sum Score, BI Barthel Index, WMFT–FAS: functional ability scale of the Wolf Motor Function Test, WMFT–TIME: performance time of the Wolf Motor Function Test, NIHSS: The National Institutes of Health Stroke Scale, MEP: motor evoked potential, UL-FMA: upper-extremity Fugl-Meyer assessment, FIM: functional independence measure, ROM: Range of motion, SSQOL: Stroke specific quality of life, SSQOL-UL = Stroke specific quality of life-Upper limb, mJTHFT: modified Jebsen-Taylor Hand Function Test, CBF: cerebral blood flow, ARAT: Action Research Arm Test, BRS: Brunnstrom stage, MI: motricity index, MFT: Manual Function Test.

## Data Availability

Not applicable.

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
