# Peer review of "Rehabilitation Interventions Combined with Noninvasive Brain Stimulation on Upper Limb Motor Function in Stroke Patients"

_brainsci, 2022, doi:10.3390/brainsci12080994_

Round 1

Reviewer 1 Report

In this systemic review, the authors present the results of 12 studies that have studied the effects of rehabilitation interventions combined with noninvasive brain stimulation on upper limb motor function in stroke patients. While the article is clearly written, the discussion is rather short and does not provide several details regarding the distinct studies and comparisons amongst them. The overall conclusion and tone in the discussion of the review leave the reader desiring more information. Yes, it is clear that noninvasive brain stimulation needs to be paired with some sort of therapy (whether a task-oriented approach or occupational therapy) to boost the effects of stimulation, but this is a conclusion that has been mentioned in previous works. Rather, it would be nice for the authors to add a sort of "future directions/considerations section", that includes what can be done to further improve rehabilitation treatment interventions. For instance, should other techniques be considered (tACS)? Should other physiological and/or imagining measures be implemented (TMS-EEG, fMRI) in clinical designs? Is there a need for individualized treatment/stimulation? What is the ideal site to target (e.g. M1 vs cerebellum)? I think the addition of a section that discusses these details, along with more information regarding the reviewed studies is needed to make this review complete.

Author Response

Dear. reviewer

I've edited it based on your advice.

Thank you for your review.

Reviewer 2 Report

The authors have reviewed 12 studies conducted within the last 10 years to find the effectiveness of rehabilitation interventions combined with NIBS or rTMS or tDCS. All of the 12 studies were randomized controlled trials. The authors review show the effectiveness of rehabilitation treatments combined with NIBS on stroke patients.

I think that the review contributes to the field of rehabilitation and neuroscience. It is publishable. I just have a minor comment:

1. line 409: the reference format

Author Response

(The authors gave the same response as above.)

Round 2

Reviewer 1 Report

The authors addressed my concerns